# Artificial Intelligence as a Diagnostic Tool in Non-Invasive Imaging in the Assessment of Coronary Artery Disease

**DOI:** 10.3390/medsci11010020

**Published:** 2023-02-24

**Authors:** Gemina Doolub, Michail Mamalakis, Samer Alabed, Rob J. Van der Geest, Andrew J. Swift, Jonathan C. L. Rodrigues, Pankaj Garg, Nikhil V. Joshi, Amardeep Dastidar

**Affiliations:** 1Translational Health Sciences, Bristol Medical School, University of Bristol, Bristol BS8 1TH, UK; 2Department of Cardiology, Bristol Heart Institute, Bristol BS2 8ED, UK; 3Department of Infection, Immunity and Cardiovascular Disease, The University of Sheffield, Sheffield S10 2TN, UK; 4INSIGNEO Institute for in Silico Medicine, The University of Sheffield, Sheffield S10 2TN, UK; 5Department of Radiology, Leiden University Medical Centre, 2333 ZA Leiden, The Netherlands; 6Department of Radiology, Royal United Hospitals, Bath BA1 3NG, UK; 7Department of Health, University of Bath, Bath BA2 7AY, UK; 8Norwich Medical School, University of East Anglia, Norwich NR4 7TJ, UK; 9Department of Cardiology, Southmead Hospital, North Bristol NHS Trust, Bristol BS10 5NB, UK

**Keywords:** artificial intelligence, coronary artery disease, cardiac imaging

## Abstract

Coronary artery disease (CAD) remains a leading cause of mortality and morbidity worldwide, and it is associated with considerable economic burden. In an ageing, multimorbid population, it has become increasingly important to develop reliable, consistent, low-risk, non-invasive means of diagnosing CAD. The evolution of multiple cardiac modalities in this field has addressed this dilemma to a large extent, not only in providing information regarding anatomical disease, as is the case with coronary computed tomography angiography (CCTA), but also in contributing critical details about functional assessment, for instance, using stress cardiac magnetic resonance (S-CMR). The field of artificial intelligence (AI) is developing at an astounding pace, especially in healthcare. In healthcare, key milestones have been achieved using AI and machine learning (ML) in various clinical settings, from smartwatches detecting arrhythmias to retinal image analysis and skin cancer prediction. In recent times, we have seen an emerging interest in developing AI-based technology in the field of cardiovascular imaging, as it is felt that ML methods have potential to overcome some limitations of current risk models by applying computer algorithms to large databases with multidimensional variables, thus enabling the inclusion of complex relationships to predict outcomes. In this paper, we review the current literature on the various applications of AI in the assessment of CAD, with a focus on multimodality imaging, followed by a discussion on future perspectives and critical challenges that this field is likely to encounter as it continues to evolve in cardiology.

## 1. Introduction

Artificial Intelligence (AI) is a general term that encompasses any computerised programme that simulates characteristics of the human intellect, such as problem-solving and learning. The field of AI is developing rapidly, especially in healthcare. For example, a current Google search for “AI in health and social care” reaps 351 million hits [1]. Current examples of the literature include the Royal College of Physicians (RCP) official statement on AI in 2018, urging the industry to address real-world challenges, doctors to scrutinise the technology, and regulators to improve guidance and evaluation methods to assess AI [2]. Eric Topol (a pioneer of individualised medicine), in his book *Deep Medicine: How Artificial Intelligence Can Make Healthcare Human Again*, explains how pattern recognition and machine learning can be used by doctors to manage health better and improve patient safety through home monitoring [3]. Topol also outlines how the connection between patients and doctors can be improved by enabling automated tasks, and thereby freeing medical professionals to focus on providing care to patients.

The modern history of AI begins in the 1950s. Alan Turing (1956) published a landmark paper in which he proposed that it would be possible to create a machine that thinks [4]. The term AI was first used by John McCarthy in 1956 at a Dartmouth conference with the theme, ‘every aspect of learning or any other feature of intelligence can be so precisely described that a machine can be made to simulate it’ [5]. The field of AI comprises machine learning (ML), which is concerned with the automated discovery of statistical patterns in data without using explicit instructions. Two important criteria for ML to function are that data is detailed enough to answer the question being asked; and also that the ML technique is appropriate for the type, amount, and complexity of the information available [6]. There are various ways in which an algorithm can model a problem based on its interaction with the experience or the environment it is merged into. The two main learning styles used in machine learning are (Figure 1):

a. Supervised learning. Input data has a known ‘ground truth’ and is labelled, e.g., cardiac images with expert hand-drawn contours, and the model is prepared through a training process in which it is required to make predictions and is corrected when those predictions are flawed. This training process goes on until the model reaches an optimal level of accuracy on the training data [7]. A practical application of supervised learning is weather forecasting for accurate weather prediction. In cardiology, supervised learning techniques such as regression models have been used traditionally in risk prediction, such as predicting the 30-day mortality risk for patients with ST-elevation myocardial infarction [8]. In cardiac imaging, automated segmentation of endocardial borders in unseen datasets is a common design goal of ML in cardiac MRI (CMR) [9].

b. Unsupervised learning. Input data is not labelled and does not have a known result. The model is prepared by deducing structures present within the input data. This may be through a mathematical process to systematically reduce redundancy, or by organising data by similarity [7]. Cluster analysis is an unsupervised ML technique which provides a process of crafting homogeneous groups of data from hidden patterns in data without prior knowledge [6]. Examples of real-world applications of unsupervised learning include genetics, where clustering DNA patterns are used to analyse evolutionary biology [10]. In cardiac imaging, clustering has been used in the echocardiographic assessment of left ventricular function [6]. Deep learning (DL) is a subtype of unsupervised ML that uses artificial neural networks (ANNs) with multiple layers to learn directly from data. As can be inferred from the name, ANNs mimic the human brain, replicating the way that neurons communicate with one another. ANNs consist of node (artificial neurons) layers that comprise an input layer, one or multiple hidden layers, and an output layer [11]. Each node is connected to another and has a weight and threshold. Once the output of any individual node exceeds the pre-determined threshold, that node is set into motion, transferring information to the next layer of the network [11]. In DL, expert hand-drawn contours are not required upfront. A convolutional neural network is an example of a DL network often used for image analysis tasks.

Significant milestones have been achieved using AI in various clinical fields, such as ophthalmology (retinal imaging), pathology (image analysis), and dermatology (differentiation of benign vs malignant skin lesions). In cardiology, AI technologies are being applied in precision medicine, clinical prediction, cardiac imaging analysis, and robotic medicine [12]. For instance, smartwatches are now being used in certain settings for arrhythmia detection, although this particular field still requires evidence and improvement.

Coronary artery disease (CAD) represents the most common form of cardiovascular disease and remains the biggest killer worldwide. For example, in the UK alone, more than 100,000 hospital admissions are attributed to myocardial infarction each year [13]. In the nature of its complexity, multiple variables, and the large number of patients it affects annually, the field of coronary disease represents an ongoing minefield for AI development and is likely to be a key area where the impact of AI will be felt. Previous studies such as Maragna et al. [14] and Gautam et al. [15] have discussed the applications of AI in CAD, including an overview of multimodality cardiac imaging, with a discussion surrounding the strengths and pitfalls of this new technology. In this paper, we aim to review the current literature on applications of AI in diagnosing CAD, with an emphasis on non-invasive cardiac imaging modalities, and present a discussion on the use of AI and MRI in prognostication. We then go on to discuss the clinical context of AI as a digital diagnostic tool; followed by a commentary on the cost-benefit implications of AI implementation in healthcare; future perspectives; and the potential impact of AI in cardiovascular imaging, including its limitations.

## 2. Echocardiography

Echocardiography remains one of the most readily available and widely used diagnostic tools in cardiology, and although the use of AI in echocardiography is still a work in progress, several applications have been developed. For instance, in their study, Madani et al. [16] trained a convolutional neural network (CNN) to recognise 15 standard echocardiographic views, using a training set of 200,000 images. The researchers demonstrated an accuracy of 91.7% compared to 79.4% for board-certified echocardiographers classifying a subset of the same test images. A possible explanation for this discrepancy offered by the authors is that the occasional misclassifications of single images most often involved views that can look similar to human eyes. These include adjacent views in the echocardiographic acquisition, where a slight difference in the angle of the sonographer’s wrist can change the view, resulting in confusion between an apical three-chamber view and an apical two-chamber view.

In addition to accuracy, other studies such as Knackstedt et al. [17] have shown that AI could enable reproducible analysis of left ventricular function, as well as longitudinal strain in approximately 8 s. Such rapid assessments of high-volume images in echocardiography could potentially save clinician time and in the process, increase the availability of echocardiogram appointments for patients, the caveat here being the reproducibility and accuracy of AI-driven assessments. Furthermore, ML confers added benefits in the evaluation of diastolic function, with a study by Lancaster et al. [18] demonstrating natural patterns of clustering of echocardiographic variables that can identify high-risk patients. Another study by Pandey et al. [19] reported that a deep neural network (DeepNN) model complemented traditional echocardiography-based risk assessment in patients with heart failure with preserved ejection fraction (HFpEF), thus potentially identifying patients most likely to benefit from spironolactone diuretic therapy. Although echocardiography has been used primarily in the past for valvular and systolic function assessments, new automated image processing technology [20] has been developed in recent times to extract various image features from stress echocardiography. Upton et al. [20] used the extracted features to train an ensemble ML classifier and found that the AI model could reduce variability and improve clinicians’ accuracy when reading stress echocardiograms, while also being able to distinguish between patients who may need revascularisation from those best treated medically, as they are less likely to have severe CAD on angiography [20].

## 3. Coronary CT

Coronary CT is a well-established modality, and due to the wealth of pre-existing data and well-validated techniques, it has been at the forefront of technology innovation and AI model development in cardiac imaging. In the CONFIRM registry, a large 5-year multi-centre prospective registry analysis [21] involving 10,030 patients undergoing coronary computed tomographic angiography (CCTA), ML was used to combine clinical and CCTA data (including segment stenosis score (SSS), segment involvement score (SIS), modified Duke index (DI), and Framingham risk score (FRS)]. In this study, 44 CCTA parameters and 25 clinical parameters were used. Predictive classifiers for all-cause mortality were developed by a classification ‘boosting’ approach, employing an interactive Logit-Boost algorithm using decision stumps for each feature-selected variable as base classifiers. The performance and general error estimation of the entire ML process was assessed using stratified 10-fold cross-validation, which is currently the preferred technique in data mining [21]. In this study, ML was found to predict 5-year all-cause mortality (ACM) significantly better than existing clinical or CCTA metrics alone (areas-under-the-curve for 5-year ACM: ML: 0.79, FRS: 0.61, SSS: 0.64, SIS: 0.64, DI: 0.62; *p* < 0.001). An important limitation of this study was that although prospectively collected, the CONFIRM registry data used to derive the model was observational and subject to selection bias and validation. Furthermore, the description of the actual ML process in this paper was rather complicated, and it represented a real challenge to readers in terms of understanding the actual methods employed.

In another study [22], 6814 asymptomatic patients underwent coronary artery calcium scoring (CAC) as part of MESA (multi-ethnic study of atherosclerosis). In this study, ML was used to process all available clinical and CT data, including the CAC score and CAC volume scores, as well as extracardiac CAC scores. Areas under the curves (AUC) by receiver operator curves were compared between clinical data alone, CAC Agatston scores alone, and a combination of all clinical and CT variables by ML. The study reported that ML of all available clinical and non-contrast CT variables was superior to clinical risk factors and the CAC score in predicting both coronary heart disease and cardiovascular disease events [22].

### 3.1. Role of Coronary Computed Tomographic Angiography (CCTA) in Prognostication

The prediction of adverse events and stratification of risk in patients using CCTA remains an area of particular interest, as it has the power to influence the specific management of these patients, and guides treatment in terms of invasive vs. non-invasive therapy, as well as appropriate pharmacotherapy. DL algorithms can undertake automated assessments of prognostic biomarkers from image data, and AI-based imaging parameters can be combined with clinical information, such as comorbidities and troponin enzyme assays, to allow risk prediction in different classes of patients [23].

### 3.2. Fat Attenuation Index

Atherosclerosis is established as an ongoing inflammatory process driving the evolution of plaque. Recent translational research on the interaction between coronary arteries and perivascular adipose tissue (PVAT) has identified the latter as an in-vivo marker of coronary inflammation by altering its composition, through the release of adipokines [24,25]. As its name suggests, PVAT is the layer of adipose tissue surrounding coronary arteries; it comprises adipocytes, stromal cells, and interstitial tissue [24,26]. The CT fat attenuation index (FAI), a metric first developed by Antonopoulos et al. [27], perceives coronary inflammation by assessing dynamic spatial changes in attenuation in PVAT, which indicates the inflammatory burden of the adjacent vessel wall [24,28]. The advantages conferred by FAI include the following: it is a sensitive, specific, and dynamic biomarker of coronary inflammation; it is independent of the severity of both coronary calcification and systemic inflammation found in the patient; and it is not confounded by the degree of coronary calcification [24,28,29]. In the landmark CRISP-CT (Cardiovascular Risk Prediction using Computed Tomography) study, the prognosis benefit of FAI was inhibited amongst patients taking primary prevention therapy with aspirin and statin following CCTA, which indicates that the risk highlighted by this marker may be modified [24,29]. Furthermore, on subgroup analysis, it was shown that FAI retained its predictive value for both cardiac and all-cause mortality, regardless of the indication for CTCA and whether chest pain was present or not [30,31] This highlights the predictive value of FAI as a risk predictor in a broader population, including asymptomatic patients [31].

In a study by Oikonomou et al. [32], the researchers designed an AI-powered technique to predict risk by analysing the CCTA-derived radiomic profile of coronary PVAT. Cases of patients experiencing major adverse cardiovascular events (MACE) within 5 years of their CCTA were selected (n = 101), as well as matched controls (n = 101), and this cohort was randomly divided into a training and testing subset, in which a random forest algorithm was used to distinguish MACE from non-MACE cases [32]. The product of the model was taken as the fat radiomic profile (FRP). This profile was then tested on 1575 participants from SCOT-HEART [33], where it was reported to considerably improve prediction for cardiac risk, beyond the current state-of-the-art methods [32]. Unlike FAI, which is a dynamic marker and can change according to responsiveness to therapy, FRP perceives adverse persistent coronary PVAT remodelling, and is thus not influenced by acute processes or medical therapy [32].

### 3.3. Plaque Feature and Fractional Flow Reserve-CT (CT-FFR)

In addition to highlighting the presence and anatomical severity of obstructive coronary disease, CCTA can identify coronary lesions with high-risk features, such as positive remodelling and low attenuation [34,35]. In fact, non-calcified plaques with densities ≤30 of Hounsfield units identified by CCTA have been reported to correlate closely with necrotic cores demonstrated in atherosclerotic plaques on intravascular ultrasound (IVUS) [36]. Studies such as Al’Aref et al. [37] have used CCTA-derived plaque features within a cohort of patients who subsequently went on to have an ACS, and designed an AI-based model for the prediction of culprit lesions from non-culprit lesions on CCTA. In the EMERALD (Exploring the Mechanism of Plaque Rupture in Acute Coronary Syndrome Using Coronary CT Angiography and Computational Fluid Dynamics) study, Lee et al. [38] sought to investigate the use of non-invasive haemodynamic evaluation to identify high-risk plaques causing subsequent ACS. In 72 patients with clear ACS and CCTA data, the presence of adverse plaque features in culprit vs. non-culprit lesions was assessed and haemodynamic parameters such as fractional flow reserve-CT (FFR-CT), change in FFR-CT (∆ CT-FFR) across the lesion, and wall shear stress (WSS) were analysed using computational fluid dynamics [38]. The authors reported that culprit lesions had worse haemodynamic parameters, and that detailed assessment with anatomical severity, adverse plaque characteristics, and axial plaque stress demonstrated discriminatory ability in identifying culprit lesions for subsequent ACS, compared with traditional anatomy-based models. In another study, Dey et al. [39] investigated whether lesion-specific ischemia by invasive FFR could be predicted by an ML-based ischemia risk score derived from plaque measurement from CCTA, using a boosted ensemble algorithm. The authors found that the new ML-powered integrated ischemia risk score showed higher prediction of ischemia, when compared to traditional individual CCTA metrics such as plaque volume or pre-test likelihood of CAD [39]. Finally, there is a growing interest in the use of CCTA and AI to guide clinical response following treatment such as PCI for significant lesions. The FFR-CT Planner [40] is a new tool that recomputes FFR-CT values after coronary angioplasty, which it achieves by combining the results of multiple simulations and reduced order modelling to calculate instantaneous FFR-CT values in a particular coronary lumen. It enables pre-procedural planning with regards to virtual stenting of coronary lesions and prediction of FFR following PCI [41]. In Sonck et al. [41], this technology was tested on 120 patients, and the authors demonstrated high accuracy and precision of the FFR-CT Planner in predicting FFR after PCI, independent of whether lesions were focal or diffuse (measured FFR post-PCI = 0.88 ± 0.06, FFRCT Planner FFR = 0.86 ± 0.06). The impact of this technology is two-fold: it can assist in personalised interventions and, at the same time, enhance patient selection, avoiding unnecessary invasive treatment in patients predicted to have little overall benefit from PCI [41].

## 4. Myocardial Perfusion Imaging

Despite the rapid emergence and relative advantages of newer techniques such as the stress cardiac MRI, nuclear cardiology remains an important non-invasive tool in the assessment of myocardial perfusion. In a registry study of 1638 patients without known CAD undergoing stress myocardial perfusion imaging (MPI), DL was trained using raw and quantitative polar maps and evaluated for prediction of clinically significant stenosis in a stratified 10-fold cross-validation procedure.

DL was shown to improve the automatic prediction of obstructive CAD, as compared to the current method, which is a parameter combining defect extent and severity to quantify hypoperfusion, known as total perfusion deficit, TPD [42,43]. There was a higher area under the receiver-operating characteristic curve (AUC) for disease prediction by DL as compared to TPD (per patient: 0.80 vs. 0.78; per vessel: 0.76 vs. 0.73: *p* < 0.01) [43]. However, an important limitation of this study was that due to the unavailability of fractional flow reserve (FFR) measurements in this population, visual stenosis on invasive coronary angiography (ICA) was used as the gold standard, which is known to overestimate the prevalence of functionally significant disease when compared to FFR studies.

In another large study by Arsanjani et al. [44], 1181 rest-stress Tc-sestamibi dual isotope MPI studies were examined. The authors found that computational integration of quantitative image measures and clinical data by ML improves the diagnostic performance of automatic MPI analysis to the level of rivalling expert analysis. A later study by Arsanjani [45] sought to investigate if early revascularisation in patients with suspected CAD could be effectively predicted by integrating clinical data and quantitative features derived from MPS by an ML approach and found that an ML-based approach was comparable or better than experienced readers in predicting early revascularisation after MPI, and that an ML approach was significantly better than standalone measures of perfusion derived from MPI.

## 5. Cardiac MRI

As a rapidly evolving imaging modality, cardiac MRI (CMR) has undoubtedly become a powerful prognostic and therapy decision tool because it provides a wealth of quantitative information on cardio-physiological parameters, tissue characterisation, and anatomical structure. This highly reliable, non-ionising modality is particularly helpful in the analysis of cardiac function and morphology and has a wide range of applications in cardiology, including the assessment of cardiomyopathies, infarction, myocarditis, and valvular heart disease, as well as congenital heart conditions. In the next section, we will look at the use of CMR in delineating cardiac scar, which has an essential role in the diagnosis as well as the management of cardiac diseases.

### 5.1. Supervised and Unsupervised Techniques for Automatic Cardiac Scar Segmentation

#### 5.1.1. Cardiac Scar Tissue Physiology

A normal heartbeat arises from spontaneous electrical activity in the heart’s natural pacemaker. This pacemaker is located in the right atrium. Electrical activity spreads through the atria and then the cardiac conduction system into the ventricles. This electrical activation acts to initiate mechanical contraction. Abnormalities of the excitation sequence are arrhythmias, which can be deadly. Arrhythmias are defined based on their location, such as atrial fibrillation, ventricular fibrillation, and supra-ventricular tachycardia. Ventricular arrhythmia (VA) is an important cause of mortality and sudden cardiac death (75–80% of cases) [46,47].

Ventricular scar is the main substrate for re-entry arrhythmias and can cause VA. Ischaemic pattern replacement fibrosis scars are created in myocardial tissue during myocardial infarction, resulting in inadequate blood supply to cardiac muscle. The slow conduction of electrical activation inside and around scars can thus generate re-entry. Re-entrant activation usually has exit sites along the border zone of the scar. Hence, the spatial distribution of myocardial scars for the accurate treatment of VA (especially post-infraction) is of major importance [48,49]. Valid and accurate mapping of the scars is crucial for correct guidance and treatment of ventricular tachycardias (VT) in catheter ablation (CA). Moreover, anatomical meshes based on realistic mapping of a scar provide an important contribution about the mechanism of cardiac arrhythmias [50]. Figure 2 [51] describes an approach used to extract 3D anatomical models of scar, as well as healthy myocardium, of the left ventricle. These anatomical models can be used to assist clinicians in evaluating and analysing the results following typical radiofrequence ablation procedures. Moreover, the 3D models can be further used to study biomarkers and mechanisms, which have the potential to increase the prognosis and success of VT treatment.

#### 5.1.2. Automatic Segmentation

Segmentation of scar tissue found in the left and right ventricle remains a challenging topic, as the huge variability of internal (e.g., size heterogeneity of scars, spatial distribution, intensity distribution) and external (e.g., resolution, noise) factors cannot be addressed by a simple model [52]. Manual segmentation of scar based on a threshold definition is a typical technique that relies on sketching contours slice by slice using pointing devices such as a mouse or trackball. Three-dimensional MRI can include as many as 100 slices. Manual segmentation can therefore be a time-consuming, tedious process which is limited by inter-observer and intra-observation variability. When only one expert is involved, the segmentation can be biased and affect the reproducibility of the results [53,54].

Automatic segmentation can overcome the limitations of manual methods. There are numerous studies in the literature that have evaluated automatic scar segmentation both in atrial regions [55,56,57,58] and ventricular regions [59,60,61,62]. Typical automated segmentation techniques refer mainly to clustering techniques such as support vector machines, c-mean, deformable methods, and other machine learning techniques. The main drawback of these methods is that the results are highly dependent on the signal-to-noise ratio of the image [63]. As a result, these algorithms frequently suffer low accuracy and robustness. Other methods are model-based methods like atlas models, which need a large dataset to capture the geometrical shape distribution of the heart for sufficient generalization [64]. Figure 3 [51,65] illustrates the role of ML in left ventricular segmentation techniques used in CMR.

#### 5.1.3. Review of Scar Automatic Segmentation Techniques

Segmentation of cardiac scar and surrounding regions, often known as the *grey zone*, can prove challenging, because of unpredictable variations in size, shape, and location. Detsky et al. [55] evaluated a multi-layer K-means clustering technique for scar segmentation of the left atrium. Their proposed automatic method was speedier and comparable to manual techniques in terms of accuracy. In their study, Lu et al. [59] utilized a graph cut segmentation technique to separate the scar region from the healthy myocardial region of the left ventricle. The main advantages of this automatic method were increased speed compared to manual segmentation and automatic detection of the scar region, as well as the prevention of false detection of scar (which is common in manual methods). The limitation of this study was the need of manual correction of 66/136 images, due to misalignment deformation between the delayed enhancement MRI. Tao et al. [52] evaluated a segmentation method which combined intensity and spatial information, for segments of unhealthy left ventricular tissue. Their approach was based on the density model theory. One assumption was that healthy tissue follows a Rician distribution [66] whilst unhealthy tissue follows a Gaussian distribution. The authors recommended this method for accurate delineation of myocardial scar, suggesting this could be a useful tool for quantitative assessment of MS in late gadolinium enhancement (LGE) MRI. In their paper, Mamalakis et al. [65] described a novel AI tool which combines atlas techniques and different traditional computer vision approaches (including k-means, active contours, mixture models, watershed etc.) in an effort to segment healthy myocardium and the scar regions of the left ventricle. They used unsupervised techniques, validating their results in an external internal unbiased ground truth (intra-observation and inter-observer ground truths).

There is an increasing trend to use DL networks for human organs automatic segmentation. The results of supervised methods tend to be better than those obtained using machine learning and computer vision algorithms. In the Medical Image Computing and Computer Assisted Interventions (MICCAI) conference in 2017, 90% of segmentation methods were based on DL [67]. In the last decade, there has been a surge in use of DL networks in the field of automated scar segmentation [68,69]. One of the main drawbacks of DL techniques is overfitting to training data. As a result, the model fits the noise of the training set and loses the generalization of the model, which can be robust to variation in vendors and cardiac shape-structure. In order to achieve valid generalization in the wider population, a large, labelled training set will be required in the future that comprises a broad spectrum of patient characteristics (such as abnormal or healthy cases, sex, age); however, obtaining this kind of dataset can prove difficult in the real world.

### 5.2. Cardiac MRI in Prognostication

In the field of cardiac MRI, machine learning techniques have mainly focused on harnessing this information through segmentation of cardiac chambers in a process that has matured over the recent years and entered clinical practice [70]. Commercially available software packages that automatically calculate CMR parameters including ejection fraction and volumes have shown robust results compared to manual assessments and proven to be powerful predictors of major adverse cardiac events in post-myocardial infarction (MI) assessment [71]. Automatic segmentation of myocardial regions of late gadolinium enhancement has also enabled the identification and quantification of ischaemic scar and shown to be predictive of clinical outcome [71]. Even without gadolinium enhancement, algorithms based on texture analysis techniques have shown an ability to detect areas of infarction. These algorithms were trained to detect scar on unenhanced CMR cine images in areas corresponding to late enhancement, enabling scar detection in situations that contradict the use of gadolinium contrast agents [72]. In addition, the severity of the ischaemic injury and scar size can be predicted using myocardial T1-mapping and myocardial strain analysis techniques [73]. In ischaemic heart disease, elevated T1-mapping values indicate areas of fibrosis or oedema [74], whilst abnormal strain-analysis usually quantifies myocardial deformation [75]. The development of DL approaches to measuring native and post-contrast values T1-mapping values [76,77] and automating strain analysis [78] (and might therefore play a role in the prognostication of ischaemic heart disease, and studies confirming this are warranted.

ML applications that extract novel CMR data beyond segmentation are still rare. Recent studies have extracted disease-specific features [79] and assessed ventricular motion throughout the cardiac cycle [80,81]. Although these approaches have been studied on pulmonary hypertension, they could be adopted and applied to CAD. In a study presented in 2019 at Euro-CMR, Dr. Aung et al. assessed the usefulness of DL and AI in the measurement of ventricular volumes, mass, and ejection fraction (six different morphotypes). The study demonstrated that DL could significantly shorten post-processing time (from 5000 studies in 7 months to 15,000 studies in <1 week). Also, 28 genes (including titin and BAG3) were identified that can predict left ventricular volumes, mass, and ejection fraction. Aug et al. also created a model to predict the risk of developing heart failure based on genetic information, sex, height, body mass index, blood pressure, dyslipidaemia, and tobacco and alcohol abuse [82] (Table 1).

## 6. Clinical Context of AI Application as a Digital Diagnostic Tool

With an ageing, multimorbid population and increasing prevalence of CAD, early exclusion of coronary disease will become increasingly important. In the assessment of CAD, the relevant questions that AI can help answer are the following:

a. Which patients should be tested? b. What tests should be requested? c. What is the timeframe for requesting the test?

The key challenge in answering these questions is balancing unnecessary, superfluous testing with optimal diagnostic accuracy of the AI model, which involves finding an equilibrium between the sensitivity and specificity of the model [83]. In a study by Overmars et al. [84], researchers trained ML algorithms to progressively exclude CAD on CCTA and CMR/SPECT in patients with chest pain, using a Dutch database, with the aim to maximize the negative predictive value of the model, in order to minimize the false negative risk with adequate specificity. The authors demonstrated the additional value of variables such as haematological markers as well as electrocardiograms in excluding CAD, concluding that ML algorithms have the potential to guide and tailor clinical management for individual patients [84].

In the broader imaging context, AI is likely to have key implications in screening pathways for conditions such as retinal disease, where AI has already shown promise in classifying 2D photographs of some common diseases, and making referral recommendations on a range of sight-threatening retinal diseases [85]. Some of the biggest companies of the world, for instance Apple, have entered the market with devices for heart monitoring such as Apple watches, which encourage individuals to perform remote monitoring even when they are not ‘patients’ [86]. This is undeniably changing the way in which the public interacts with their health habits and medical professionals. AI could facilitate ‘one-stop-shop’ consultations for patients with known or unknown CAD, with large data enabling focused and rapid diagnosis and thorough patient discussion, together with specific risk stratification.

The amalgamation of AI and CAD requires technical skills, cutting-edge technologies, and considerable financial and resource investment [12]. It is likely that cardiologists will be trained in data expertise in order to understand the clinical needs and challenges. In the near future, it is highly probable that AI projects will be conducted by large technology corporations such as Apple and Microsoft. The recent Apple Watch series 4 has a transducer that enables ECG measurement, and it is only a matter of time before technological giants combine big data and DL algorithms to develop innovative devices that shape the future of coronary disease prevention and diagnosis worldwide. We have already seen in the last decade a meteoric rise in the use of individual data available on social media platforms such as Twitter and Facebook, in order to inform and manipulate behaviour. Several authors, such as Silicon Valley pioneer Jaron Lanier, have argued that social media platforms have the potential to twist our relationship with the truth and reduce autonomy by continuously prodding the public with algorithms designed to alter behaviour [87]. It will thus be vital for our society to agree on how to define the limits and responsibilities of these companies if they were to step into healthcare. Figure 4 illustrates the stakeholders in the future expansion of AI in enabling personalized medicine [88,89].

## 7. What Is the Future of AI in Cardiovascular Imaging?

There is a unique opportunity in the next few years for cardiovascular imaging to be at the forefront of the application of AI, including ML. AI introduces new possibilities in terms of reducing human error and saves time in the clinical workflow through automatic segmentation of cardiac structures in far less time than manual techniques, as discussed earlier [90]. ML has the potential to maximize the information obtained from diagnostic echocardiographic, CT, or CMR images solely or in a combination of imaging and clinical predictors, thereby enabling disease diagnosis and prognosis, but also risk stratification influencing management strategies. The use of AI in cardiovascular imaging can in the future translate into increased processing speeds, improved accuracy, higher test volumes, reduced human expert time and, in certain cases, enhanced safety to patients where, for instance, non-invasive tests such as CT-fractional flow reserve) (CT-FFR) would avoid the need for unnecessary invasive coronary angiography—the caveat being that much more work in this field still requires cross-validation and comparison with traditional methods before they can be fully and widely implemented. The next decade is also likely to bring exciting opportunities to merge data from biomarkers, epigenetics, and proteomics, alongside imaging data and clinical cardiovascular risk factors in order to refine the predictive accuracy and power of ML algorithms and thus empower physicians to deliver highly personalized healthcare to their patients.

## 8. Limitations and Challenges

One criticism of AI when used in healthcare is the challenge associated with interpreting the output of a particular ML algorithm. There is a notable difference between ‘interpretable’ and ‘explainable’ data—the former refers to ML models that can be understood by humans (for instance, decision tree models); whereas the latter pertains to overly complex models such as neural networks, which require further tools to gain an understanding of how they function [91]. Therefore, there is a steep learning curve when it comes to developing algorithms, as well as novel technology for treatment. In the future, this is likely to translate to special modules focusing on AI, and already numerous courses have been developed to equip the workforce for digital transformation. Secondly, there is no known randomized trial to date showing that improved phenotyping and personalized medicine based on this leads to better outcomes. At present, the majority of newly developed ML models are validated in single-centre studies with a restricted number of cases [14]. Early-stage evaluation studies are needed to investigate whether AI can lead to better care and outcomes whilst lowering costs [90]. However, reporting of early studies remains suboptimal. In an effort to address this issue, the DECIDE-AI guideline [92] was deigned taking into consideration international expert consensus, in order to report studies looking at the early-stage clinical assessment of AI-based decision support systems [92]. The DECIDE-AI checklists comprise information on the following: title, abstract, study objectives, research governance, participants, AI system, implementation, outcomes, safety and errors, analysis, ethics, human factors, main results, human-computer-agreement, and support for intended use [92].

One important limitation of ML at the development stage is sampling bias [15,93]. This arises because ML models are usually built from large datasets, some of which are obtained from electronic health records (EHRs). Because of the nature of these datasets, information extracted can be skewed and unrepresentative of the wider population, thus resulting in considerable bias and lack of generalizability [15]. An example of this is the underrepresentation of ethnic minorities in datasets but also clinical trials [14,94,95]. This is a crucial problem, as it can lead to these groups potentially missing out on life-saving treatment and novel technology, with adverse outcomes [14]. In fact, disparities in cardiovascular outcomes have been reported in the last few decades with regards to ethnic minorities [96,97,98]. Although some models have attempted to avoid losing representation of subgroups by stratifying their datasets, randomised trials are urgently needed to address this bias and evaluate the model’s performance against traditional clinical criteria [15]. At present, there are no clear guidelines for reporting bias in AI-based studies. Therefore, in recent times, protocols have been advocated for the development of a reporting guideline (TRIPOD-AI) and risk of bias tool (PROBAST-AI) for diagnostic and prognostic prediction model studies based on AI [99]. These are the AI versions of the TRIPOD Transparent Reporting of a multivariable prediction model of Individual Prognosis or Diagnosis) statement and the PROBAST (Prediction Model Risk of Bias Assessment Tool), and whilst they are currently works in progress, it is anticipated that PROBAST-AI will help the scientific community to critically evaluate the design, conduct, and analysis of AI-based prediction model studies, with a reliable, standardised tool for bias assessment [99].

Another considerable challenge faced by data scientists building ML models is the reliance on large datasets for model training and testing. As mentioned earlier, these large datasets are derived from EHRs. These are often subject to data heterogeneity, which arises because data is collected across different hospital sites, introducing variation in definitions and coding systems for diagnoses, as well as enzyme assays [15,100]. Another significant concern of large datasets is missing data, which can arise due to human error or system malfunction [101]. Missing values can lead to biased outcomes and reduced ML model performance, although techniques such as imputation can help reduce the problem [102]. Overfitting is an issue that arises when the model struggles with processing components of the testing dataset, which may differ from the training set. Overfitted models have a tendency to memorise the whole data, including irrelevant noise on the training set, instead of learning the pattern concealed behind the data [103]. In the medical field, more precisely in deciphering the correlation between symptoms and final diagnosis, an example of overfitting would be as follows: if in error the patient’s hospital number is entered as one input feature, an overfitted model may conclude that the illness in question is influenced by the hospital number [103,104].

There are also other factors to consider. According to a white paper by Stanford Medicine [105], the sheer volume of healthcare data is growing at an astronomical rate: an estimated 2314 exabytes (one exabyte = one billion gigabytes) would have been produced in 2020. With such vast amounts of data available it may become confusing to cardiologists and patients. As a result, clinicians will have to ensure that AI-generated findings are critically appraised when considering treatment implications for patients. Another important challenge will be navigating the evolving scientific, ethical, and regulatory landscape, including patient privacy [91,106]. Indeed, data privacy issues can arise since commercial algorithms are designed by organizations that require access to electronic health records. Thus, whenever assessing those algorithms, users and clinicians should keep informed as to who manages the source data, as well as the precautions taken to protect patient anonymity and privacy [91].

## 9. Cost-Benefit Implications

As with any innovative technology, healthcare organisations will need to be meticulous when conducting their cost-benefit assessments, in order to evaluate the potential value of the proposed novel AI-assisted cardiac imaging tool prior to implementation [107]. This will involve balanced considerations relating to direct costs involved with the new AI imaging tool, such as software pricing; information technology (IT) specialists and data scientists to refine the model, develop apps, and ensure seamless integration of the software onto the healthcare interface; data protection officers in charge of overseeing the overall data processing agreement; as well as imaging trained specialists such as radiologists, echocardiographers, and MRI-trained cardiologists who will be responsible for testing the new AI solution, providing feedback, and assisting in rolling out the model on a broader scale [108]. These costs will need to be carefully weighed against anticipated gains, such as potential cost savings per patient when factoring in faster processing times resulting in shorter patient waiting lists; percentage reduction of missed diagnoses, assuming that the new AI technology offers greater sensitivity and specificity [109]. In-depth studies regarding the cost effectiveness and related quality-adjusted life year (QALY) associated with using an AI-based intervention will be necessary to assess clinical and economic implications [110]. Various programmes have been set up recently to support the development, evaluation, and cost assessment of promising new AI technology. In fact, the accelerated Access Collaborative (AAC), in partnership with the National Institute for Health Research (NIHR), has introduced the AI in Health and Care Award [110], which is a National Health Service (NHS) AI Lab programme designed to support technologies across a spectrum of development, from initial feasibility to final evaluation, within the NHS. This award’s key role is to help establish a broad network of technology testing infrastructure for innovation [110].

## 10. Conclusions

There is no doubt that AI holds tremendous potential for developing and improving patient care, as demonstrated through ML models that can be used to improve diagnostics and risk prediction in the field of non-invasive cardiac imaging and the assessment of CAD. However, this comes with a word of caution. Despite the considerable potential of AI, it is still likely that due to the complex intricacies of each individual patient, it will still be important for the human expert to take in these large clusters of information, process them, and objectively present the pros and cons to the patient, taking into consideration patient beliefs, opinions, concerns and anxieties—in a way that no machine can. A multidisciplinary approach in tackling all facets of coronary disease will therefore remain of prime importance.

## 11. Future Perspectives: Highlights

Artificial intelligence offers huge potential in cardiac imaging and the assessment of CAD; and by reducing human error and saving considerable amounts of time, AI has the capacity to unlock rapid, personalized treatment strategies for patients.

Challenges that are likely to emerge include ethical issues and patient privacy, as well as training requirements, to adapt to new technology.

A multidisciplinary approach remains key in combining AI and human expertise to offer the best treatment to patients.

## Figures and Tables

**Figure 1 medsci-11-00020-f001:**
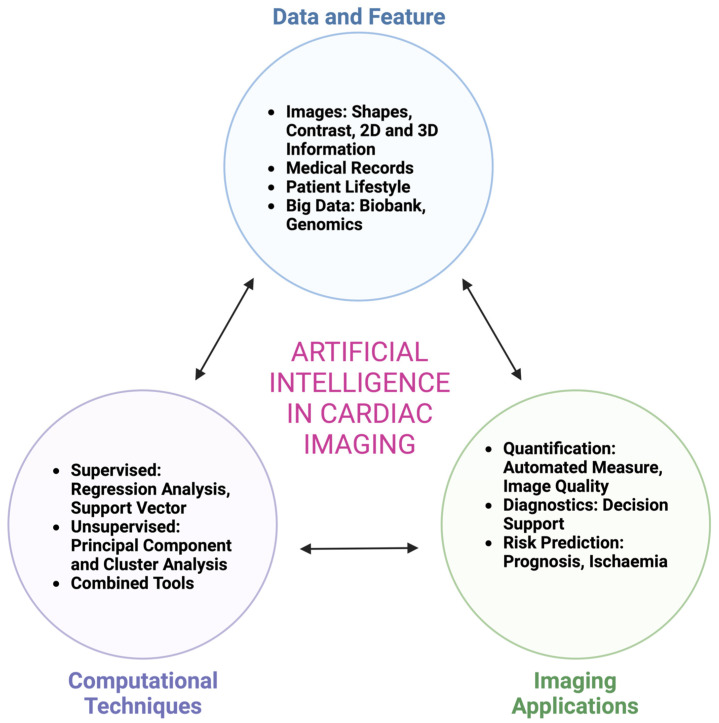
Steps to use artificial intelligence in cardiac imaging. [modified from Sey et al. [6]].

**Figure 2 medsci-11-00020-f002:**
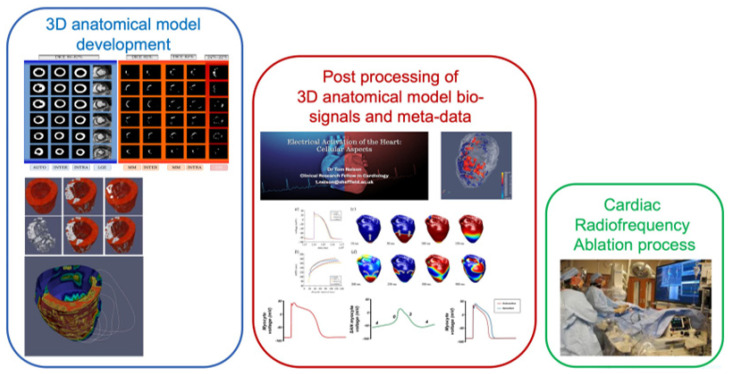
3D anatomical model framework to assess cardiac arrhythmia risk [51].

**Figure 3 medsci-11-00020-f003:**
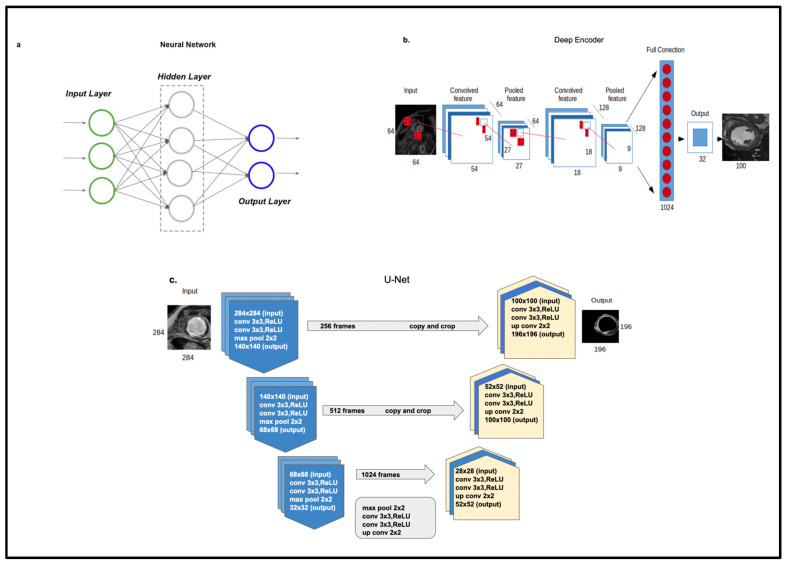
Left ventricular segmentation techniques used in CMR: (**a**). Neural network, (**b**). Deep encoder, (**c**). U-net architecture [51,65].

**Figure 4 medsci-11-00020-f004:**
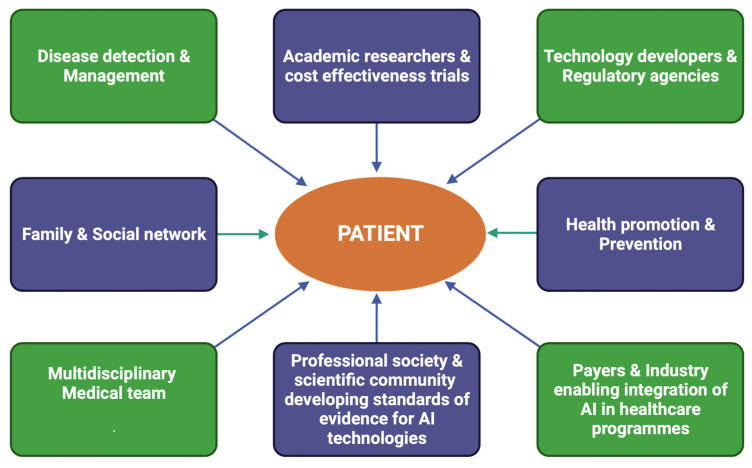
Stakeholders in the development of Al enabling personalized treatment in healthcare (modified from Mc Connell et al. & Sharma et al. [60,63]).

**Table 1 medsci-11-00020-t001:** Summary of some studies looking at the application of artificial intelligence and machine learning in non-invasive imaging for assessment of CAD.

Study Author	Year	Modality	Population	Description	Main Findings
Arsanjani et al. [20]	2013	MPI	1181	This study aimed to improve the accuracy of myocardial perfusion SPECT (MPS). 1181 MPS studies were examined, including 713 cases with correlating invasive coronary angiography data. Clinical data and quantitative image features were integrated with ML algorithms. TPD and stress/rest perfusion change were obtained from automated perfusion quantification software and combined with variables such as age and sex by LogitBoost.	Computational integration of quantitative image measures and clinical data by ML improves the diagnostic performance of automatic MPI analysis to the level rivalling expert analysis.
Knackstedt et al. [15]	2015	Echo	255	ML analysis was used for fully automated left ventricular measurements including EF, as well as longitudinal strain. A reference centre re-examined all datasets by visual estimation, as well as manual tracking.	Automated left ventricular measurements were completed in 98% of studies, with good reproducibility and an average analysis time of 8 s.
Avendi et al. [35]	2016	CMR	45	This study utilised DL algorithms combined with deformable models in order to design a fully automated left ventricular segmentation model from short-axis CMR datasets. DL was used for automatic detection and inferring left ventricular shape.	Excellent agreement and high correlation with reference contours were reported.
Dawes et al. [54]	2017	CMR	256	This study investigated whether patient survival in pulmonary hypertension (PH) could be predicted using ML of 3-D patterns of cardiac motion on CMR. All patients with new diagnosis of PH underwent CMR, right heart catheterisation, and a 6-minute walk.	The ML survival model was found to predict outcome independent of traditional risk factors in patients with newly diagnosed PH.
Motwani et al. [16]	2017	CT	10,030	This was a registry analysis of 10,030 patients with suspected CAD. 25 clinical and 44 CCTA parameters were assessed, and ML involving automated feature selection and model building with a boosted ensemble algorith, was used to combine a clinical and CCTA, modified Duke index and Framingham risk score to predict all-cause mortality.	ML was found to predict 5-year mortality significantly better than existing clinical or CCTA metrics alone.
Madani et al. [14]	2018	Echo	267	CNN was trained to recognize 15 standard echocardiographic views, using a training set of 200,000 images based on still images and videos from 267 transthoracic echocardiograms.	DL achieved expert-level classification, with researchers demonstrating an accuracy of 91.7% compared to 79.4% for board certified echocardiographers classifying a subset of the same test images.
Nakashini et al. [17]	2018	CT	6814	This study included data from 6814 asymptomatic patients undergoing coronary artery calcium scanning who were followed up for coronary heart disease and atherosclerotic cardiovascular events over a decade. ML utilised all available clinical and CT data including the CAC score, CAC volume scores, as well as extracardiac CAC scores.	ML of all available clinical and non-contrast CT variables was superior to clinical risk factors and CAC score in predicting both coronary heart disease and cardiovascular disease events.
Betancur et al. [19]	2018	MPI	1638	This study compared the automated prediction of obstructive disease from MPI by DL with total perfusion deficit (TPD). Patients without known CAD underwent stress ^99m^Tc-Sestamibi or tetrofosmin myocardial perfusion imaging (MPI). DL was trained using raw and quantitative polar maps and evaluated for prediction of clinically significant stenosis in a stratified 10-fold cross-validation procedure.	DL was shown to improve automatic prediction of obstructive CAD, as compared to the current method. AUC from the ROC curve for disease prediction by DL was higher than for TPD (per patient: 0.80 vs. 0.78; per vessel: 0.76 vs 0.73, *p* < 0.01).
Zabihollahy et al. [36]	2018	CMR	34	DL was used to design a semi-automated method for fully automated segmentation of a left ventricular scar from 3-D late gadolinium CMR images from patients with ischaemic cardiomyopathy, without any operator interaction.	The new method was found to outperform alternative techniques.
Zheng et al. [41]	2018	CMR	3078 from UK Biobank(training), 756 (testing)	DL was used to carry out cardiac segmentation with spatial propagation on CMR image stacks. The method was trained on a large database of 3078 cases and then tested on 756 cases	This technique achieved comparable and even improved results in terms of distance measures when compared with state-of-the-art methods.
Baessler et al. [46]	2018	CMR	120	This was a proof-of-concept study assessing whether texture analysis allowed for the diagnosis of subacute and chronic MI on CMR images. 120 patients undergoing CMR showing large transmural infarcts or small chronic ischaemic scars were entered retrospectively. Regions of interest for texture analysis involving the left ventricle were contoured by 2 blinded readers on cine images by using a software package. Texture feature selection based on reproducibility, ML and correlation were carried out for selecting features, allowing the diagnosis of MI on non-enhanced CMR images by using LGE as standard of reference.	The authors concluded that texture analysis enabled the diagnosis of subacute and chronic MI with high accuracy.
Fahmy et al. [50]	2019	CMR	210 (training and testing), 455 (validation)	In this study, the authors describe an automated technique (deep fully convolutional neural network, FCN) which was used for myocardial segmentation in T1 weighted CMR images.	FCN enabled fast segmentation (<0.3 s per image) with a high Dice similarity coefficient, thus allowing fast automatic analysis of myocardial native T1 mapping images on CMR.
Schuster et al. [45]	2020	CMR	1017	CMR data from 2 MI multicentre trials (n = 1017 patients) were included and analysis of parameters such as EF were manually and automatically assessed using conventional and AI-based software. Obtained measurements entered regression analysis for prediction of MACE.	Volumetric analysis carried out by AI software was feasible, with results being reported to be equally predictive of MACE compared with traditional methods.
Ferdian et al. [52]	2020	CMR	4508 from Biobank (3244 for training, 812 for validation, 452 for testing)	This was a retrospective cross-sectional study whereby neural networks (including CNN) were used to perceive and track the myocardial landmarks through each slice, and strain measurements were made from the landmarks’ motion.	The automated technique allowed unbiased strain assessment, with a typical processing time of 260 frames (13 slices) per second, compared with 6–8 min per slice for manual methods.
Swift et al. [53]	2021	CMR	220	This study investigated the use of a tensor-based ML approach to highlight features of PAH using CMR. Untreated patients with PAH or no evidence of pulmonary hypertension (PH) who underwent CMR and right heart catheterisation studies within 48 h were selected from the ASPIRE registry. A tensor-based ML model was developed, and the accuracy of this tool was measured against standard CMR assessments.	The authors reported high diagnostic accuracy as assessed by AUC at receiver operating characteristic analysis (ROC), *p* < 0.001:0.92 for PAH, which is slightly higher than standard CMR assessments.

AUC = area under curve; AI = artificial intelligence; CAC = coronary artery calcium; CAD = coronary artery disease; CCTA = coronary computed tomography angiography; CMR = cardiac magnetic resonance; CNN = convolutional neural network); DL = deep learning; EF = ejection fraction; LGE = Late gadolinium enhancement; MACE = major adverse cardiac events; MI = myocardial infarction; ML = machine learning; MPS = myocardial perfusion imaging; PAH = pulmonary arterial hypertension; PH = pulmonary hypertension.

## Data Availability

Not applicable.

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
