# Peer review of "Artificial Intelligence as a Diagnostic Tool in Non-Invasive Imaging in the Assessment of Coronary Artery Disease"

_medsci, 2023, doi:10.3390/medsci11010020_

Round 1

Reviewer 1 Report

This is a well-written review paper on the apport of artificial intelligence in coronary artery disease diagnostics. The benefits of AI are clearly stated and backed up with relevant literature. The authors should add a paragraph on the cost-benefit of this technique and its availability. The section on ultrasound can be enriched by giving more details on the benefits of using AI to assess diastolic function (e.g., Lancaster MC, Salem Omar AM, Narula S, Kulkarni H, Narula J, Sengupta PP. Phenotypic clustering of left ventricular diastolic function parameters: patterns and prognostic relevance. JACC: Cardiovascular Imaging. 2019 Jul;12(7 Part 1):1149-61.

Page 10 ln 361: ..." was" produced in 2022

Figure 3 is hard to read

Reviewer 2 Report

The manuscript "Artificial Intelligence as a Diagnostic Tool in Non-Invasive Imaging in the Assessment of Coronary Artery Disease" reviews the application of artificial intelligence (AI) in cardiology, in particular, for diagnostics of coronary artery disease (CAD). I have some major and minor concerns regarding this manuscript which I want to share with the authors.

Major concerns:

1. What was the motivation of writing this manuscript? Do the authors themselves work in the development of AI tools for CAD diagnostics?

2. What new this review article introduces if compared with the articles "Artificial Intelligence in Cardiovascular Imaging for Risk Stratification in Coronary Artery Disease" published by Lin et al. in 2021, "Artificial Intelligence Based Multimodality Imaging: A New Frontier in Coronary Artery Disease Management" published by Maragna et al. in 2021, "Current and Future Applications of Artificial Intelligence in Coronary Artery Disease" published by Gautam et al. in 2022, or any other AI for CAD diagnostics review article published recently?

3. Why the articles listed in the previous comment were not mentioned or discussed in this manuscript?

4. Writing style of this manuscript is twofold: sections 2-4 are written in academic way, while sections 1, 5-8 use more of a journalistic style with some speculations but lack of scientific facts justified by references.

5. The abstract of this manuscript repeats several sentences taken from the Introduction section. A dedicated abstract should be written.

Minor concerns:

1. It is hard to understand the meaning of the sentence in lines 150-152.

2. A reference needs to be added to Figure 2 if this figure was not created by the authors.

3. In line 322, a reference to Figure 3 is provided, but the authors in fact refer to Figure 4.

4. Figure 3 is mentioned earlier than Figure 2, so they should be swapped.

5. Figure 3 needs to be put closer to where it was mentioned in the text.

6. Abbreviations FFR and CT-FFR were not explained.

Reviewer 3 Report

Artificial intelligence techniques are increasingly applied in medical areas that require more accurate pattern recognition  for diagnosis and in predictive clinical models, due to their ability to extract meaningful relations from large datasets (Eur Heart J 2019; 0, 1–9 doi:10.1093/eurheartj/ehz565, European Heart Journal 2019, 40, 3529–3543 doi:10.1093/eurheartj/ehz592,  J Am Coll Cardiol 2019; 73:1317–35). In this  paper, Doolub G. et al aim at reviewing the use of AI as a "diagnostic tool" in non-invasive imaging assessment of CAD. However, although the issue is of great clinical relevance and interest tfor the readers of Medical Sciences, their review is neither systematic nor exhaustive and thus fails to add any significant improvements or update on this topic -  with the only exception of AI application to CNMR non invasive imaging - within the multitude of similar papers (see for example the recent systematic or descriptive reviews   Diagnostics. 2021; 11(3):551. https://doi.org/10.3390/diagnostics11030551, APL Bioeng. 5, 011505 (2021); doi: 10.1063/5.0028986 and Computers in Biology and Medicine Volume 128, January 2021, 104095) .

Furthermore, the clinical context of AI application as a digital diagnostic tool and its potential impact  is not specified.  The current clinical questions to be addressed in CAD assessment are 1. who should be investigated, 2. what test should be ordered and 3. when should it be performed: balancing minimal unnecessary overtesting and maximal diagnostic accuracy as to achieve efficacy of treatment is the main  challenge facing a strategy based on AI approaches  ( European Heart Journal Digital Health, 2021 doi 10.1093/ehjdh/ztab103 ).  

Finally , the limitations in AI application are not properly discussed. The reporting of clinical models using AI is poor and there are no guidelines for the reporting or risk of bias assessment: therefore, protocols have  been recently proposed for development of a reporting guideline (TRIPOD-AI) and risk of bias tool (PROBAST-AI) for diagnostic and prognostic prediction model studies based on AI (BMJ Open 2021;11:e048008. doi:10.1136/bmjopen-2020-048008) as well as reporting guideline for the early-stage clinical evaluation of decision support systems driven by artificial intelligence  (DECIDE-AI) (Nature Medicine | VOL 28 | May 2022 | 924–933 | www.nature.com/naturemedicine).

Reviewer 4 Report

In this paper, the authors aim to systemically review the current literature on applications of AI in diagnosing CAD, with an emphasis on non-invasive imaging modalities, followed by a discussion on future perspectives and challenges that this field may encounter as it continues to evolve in cardiology.

Is underlined as there is no doubt that AI holds tremendous potential for developing and improving patient care, as demonstrated through ML models that can be used to improve diagnostics and risk prediction in the field of non-invasive cardiac imaging and the assessment of CAD. However, this comes with a word of caution. Despite the considerable potential of AI, it is still likely that due to the complex intricacies of each individual patient, it will still be important for the human expert to take in these large clusters of information, process them, and objectively present the pros and cons to the patient, taking into consideration patient beliefs, opinions, concerns and anxieties- in a way that no machine can. A multidisciplinary approach in tackling all facets of coronary disease will therefore remain of prime importance.

The article turns out to be well-written and enriched with a lot of data and bibliography that make the topic being discussed decisive and high-impact.

I recommend a complete revision of the English language to make the data covered better understandable.

Round 2

Reviewer 2 Report

The manuscript was improved well and I recommend accepting it for publishing in Medical Sciences.

Reviewer 3 Report

We acknowledge that the authors revised the manuscript according to suggestions and their descriptive  review of AI application to non-invasive coronary imaging is now more exhaustive.